# In Vivo Optogenetic Modulation with Simultaneous Neural Detection Using Microelectrode Array Integrated with Optical Fiber

**DOI:** 10.3390/s20164526

**Published:** 2020-08-13

**Authors:** Penghui Fan, Yilin Song, Shengwei Xu, Yuchuan Dai, Yiding Wang, Botao Lu, Jingyu Xie, Hao Wang, Xinxia Cai

**Affiliations:** 1Aerospace Information Research Institute Chinese Academy of Sciences, Beijing 100190, China; fan_ph@163.com (P.F.); ylsong@mail.ie.ac.cn (Y.S.); swxu@mail.ie.ac.cn (S.X.); hongri1991@126.com (Y.D.); wangyiding18@mails.ucas.ac.cn (Y.W.); lslbt96@163.com (B.L.); xiejingyu16@mails.ucas.ac.cn (J.X.); wanghao161@mails.ucas.ac.cn (H.W.); 2University of the Chinese Academy of Sciences, Beijing 100049, China

**Keywords:** optogenetics, micro-electrode array, in situ detection, electrophysiology, neural circuit recognition

## Abstract

The detection of neuroelectrophysiology while performing optogenetic modulation can provide more reliable and useful information for neural research. In this study, an optical fiber and a microelectrode array were integrated through hot-melt adhesive bonding, which combined optogenetics and electrophysiological detection technology to achieve neuromodulation and neuronal activity recording. We carried out the experiments on the activation and electrophysiological detection of infected neurons at the depth range of 900–1250 μm in the brain which covers hippocampal CA1 and a part of the upper cortical area, analyzed a possible local inhibition circuit by combining opotogenetic modulation and electrophysiological characteristics and explored the effects of different optical patterns and light powers on the neuromodulation. It was found that optogenetics, combined with neural recording technology, could provide more information and ideas for neural circuit recognition. In this study, the optical stimulation with low frequency and large duty cycle induces more intense neuronal activity and larger light power induced more action potentials of neurons within a certain power range (1.032 mW–1.584 mW). The present study provided an efficient method for the detection and modulation of neurons in vivo and an effective tool to study neural circuit in the brain.

## 1. Introduction

Optogenetic technology has been widely used in the research of behavior, diseases and other aspects of neural mechanism since it was put forward in 2005 [1], and not only on the central nervous system; it also had effects on the peripheral nervous system [2,3]. Optogenetic technology can modulate the activities of target neuron cells with high spatial-temporal resolution by combining photonics and genetic techniques and has been one of the most important neuroscience research tools [4,5,6]. The core of optogenetics is light-sensitive protein, also called opsins, which are mainly found in micro-organisms. The opsin gene is introduced into the subject’s brain through viral vectors, and is eventually expressed in targeted cells after transcription [7,8]. The light-sensitive protein expressed in the cell membrane acts as an ion channel or pump controlled by light in a specific wavelength range. Channelrhodopsin-2 (ChR2) is one of the most commonly used opsin tools, which promotes cell depolarization when exposed to blue light.

When optogenetic technology was used to modulate neural activity, a large amount of neuronal activity needed to be recorded, mined and utilized to help us better understand the neurophysiology of the nervous system. Optogenetics combined neuronal recording can also help to identify neuronal subtypes and circuit [9]. Therefore, a tool that can simultaneously transmit light to targeted cells and detect electrophysiological information is particularly important. Microelectrode array (MEA), based on the Micro-Electro-Mechanical System (MEMS) method, is considered to be an ideal platform to detect the neural activity of a single neuron, which also has the potential of electrochemical detection. At present, the integrated device combining optical interface and MEA, also called optrode, which can be classified according to integrated optical fiber, waveguide and micro light emitting diode (µ-LED), has attracted attention [10,11]. The optrode integrating optical fiber is a kind of mature and effective optogenetic tool [12,13]. HyungDal Park formed the V-groove on the MEA by KOH wet etching process to assist the alignment of the optical fiber and electrode [14]. Wang and colleagues bonded a cannula to the open hole on the backside of the MEA to guide the fiber [15]. This method can be successfully applied to the field of optogenetics, but requires considerable time and effort to develop. The optrode based on waveguide is completed by directly depositing the waveguide material (usually SU-8) on the MEA through the MEMS technology. Since the optical waveguide only act as optical transmission medium, it is difficult to couple the light source and the optical waveguide, which usually generates a large light loss and increases cost [16,17,18]. The μ-LED itself is a low-power light source, and can directly stimulate tissues with minimal light loss, so it is often integrated on MEA [19,20]. Furthermore, the manufacturing technology is usually forming the μ-LED structure on silicon probe through metalorganic vapor phase epitaxy (MOVPE) or directly bonding the commercial μ-LED to the MEA. However, this method is complicated and even cause tissue damage due to overheating of the light source [21,22].

Considering the cost and operability, we produced an optrode that combines optical fiber and MEA through hot-melt adhesive. We used a homemade MEA that has been connected to Printed Circuit Board (PCB) which increased the operable space and provided a larger and harder contact area for fiber bonding. The device had the characteristics of low cost, easy operation and reusability for acute in vivo experiment. Here, we verified the performance of the integrated device, analyzed a possible circuit based on the performance of neurons under the optogenetic control and explored the effect of optical patterns and light powers on neuronal activity. This study provided an efficient tool for optogenetic neuromodulation and simultaneous in vivo electrophysiological detection.

## 2. Materials and Methods

### 2.1. Fabrication and Preparation of MEA Integrated with Optical Fiber

The Silicon-based MEA was fabricated based on MEMS technology as previously reported [23], which consisted of four 6 mm shanks. Each shank was 100 μm in width, 30 μm in thickness and 80 μm spacing. There were four electrode sites (16 μm in diameter) on the tip of every shank forming a 4 × 4 array. The fabrication process is shown as Figure 1a: (1) the SiO_2_ layer was deposited on the silicon-on-insulator (SOI, 30 μm Si/2 μm SiO_2_/450 μm Si) substrate to insulate the microelectrode from the substrate; (2) then, the Pt/Ti conductive layer pattern (including recording sites, bonding pads and conductive lines) was formed by photolithography, sputtering and lift-off process; (3) the SiO_2_/Si_3_N_4_ insulating layer was deposited via plasma enhanced chemical vapor deposition (PECVD) and the recording sites and bonding pads were exposed by the second photolithography and CHF_3_ reactive ion etching (RIE); (4) the shape of the MEA was established by the third photolithography and inductively coupled plasma deep reactive ion etching (ICP DRIE); (5) the backside silicon was wet etched and the silicon probes were released by self-stop etching in KOH solution. Then, the pads of individual probe were connected to the printed circuit board (PCB) through wire bonding and embedded in silicone rubber (Nanda 705#, Liyang Kangda Chemical Co. Ltd., Liyang, China) to isolate electrical connection. Finally, the assembled MEA was electroplated with platinum nanoparticles to improve recording performance.

In order to integrate with the optical interface, a 20 mm optical fiber (200 μm in diameter, NA = 0.39, CFMLC12L20, Thorlabs, USA) was closely attached to the PCB of the assembled MEA. Then the fiber was adjusted to be as parallel as possible with probes, and the fiber tip was about 200 μm above the recording sites. Finally, hot-melt adhesive was used to bond the optical fiber and MEA to ensure mechanical stability (Figure 1b). This method was simple in operation, short in time and low in light loss. The fiber and MEA could even be separated under heating, which undoubtedly improved their utilization rates. At the same time, the integrated device could be reused for acute experiments.

### 2.2. ChR2 Transduction and Optrode Implantation 

The experimental procedures were conducted with the permission of Beijing Association on Laboratory Animal Care. The C57 mouse (7-week-old, 25 g) was anesthetized by the isoflurane anesthesia apparatus (RWD520, RWD Life Science, Shenzhen, China) and fixed in the stereotaxic frame (51, 600, Stenting, Wood Dale, IL, USA). The craniotomy was centered at AP = −2.3 mm and ML = 1.5 mm for targeting the mouse hippocampus. Then 0.5 μL adeno-associated viruses (rAAV-hSyn-hChR2(H134R) -mCherry-WPRE-hGH PolyA, BrainVTA, Wuhan, China) were injected into the mouse brain at a depth of 1 mm (Corresponding to the pyramidal cell layer of hippocampal CA1). After the injection, the scalp was sutured, and penicillin was applied to the wound to prevent infection. Then the mouse was placed on a soft cotton pad and returned to the rearing cage after being fully awake.

After 4 weeks, the integrated device was implanted into the target brain area at the same site for virus injection under anesthesia using a micropositioner (model 2662, David KOPF instrument, USA) for testing. The depth of electrophysiological signal recording is mainly concentrated from 800 μm to 1250 μm. The schematic diagram of device implantation is shown in Figure 1b.

### 2.3. Electrophysiological Recording and Optical Stimulation

For real-time electrophysiological recordings, the MEA was connected to the recordings system (USB-ME16-FAI-System, MultiChannel Systems, Reutlingen, Germany) via a 16-channel headstage. The electrophysiological signals were sampled at the rate of 25 kHz. A high pass filter (200 Hz) was used to obtain action potentials (spikes), and a low pass filter (200 Hz) was applied to obtain local field potentials (LFPs). During recording, the background noise was about ±13 μV, and the action potentials trigger threshold was set to −40 μV to ensure that the signal-to-noise ratio of the signal was greater than 3. 

For optical stimulation, the light was applied by fiber-coupled LED (M450F, 450 nm, Throlabs, Newton, NJ, USA), and the optical fiber on the optrode was coupled to the light source through a ceramic mating sleeve. We used the Labview program to drive the LED light source, and change the output power of the LED by setting the driving voltage. The maximum driving voltage was 5000 mV, which corresponded to the maximum output power of the LED. The schematic diagram of the experiment is shown in Appendix A.

### 2.4. Histology

Following completion of test, the mouse was injected with 10% chloral hydrate into deep anesthesia and perfused through the heart with 0.9% NaCl and 4% paraformaldehyde in sequence. After perfusion, the brain was removed and put into 20% sucrose solution for 24 h followed by 30% sucrose solution for 48 h for postfication. Then 40 μm thick sections were prepared with a freezing microtome and mounted on glass slides, and sections were viewed and photographed under the confocal fluorescence microscopy. The fluorescence image is shown as Figure 1c, due to the spread of the virus, the infection area covers the hippocampal CA1 and part of the cortex above CA1. The estimated spread of ChR2 was 400 μm in the vertical plane.

### 2.5. Statistics

We used a mouse in this article to verify the device performance and made repeated experiments to ensure the accuracy of the conclusion. There is enough blank time between different experiments to eliminate the interference between them. Therefore, all tests were administrated independently. Spike data were sent to Offline Sorter for sorting and clustering analyses. Spike waveforms were purified and extracted by principal component analysis (PCA). Then sorted spike data and LFP data were further analyzed by NeuroExplorer as the following:(1)Spike firing rate analysis was defined as the mean number of spikes per second each bin. Here we took the average of multiple recording channels.(2)Interspike interval (ISI) analysis was defined as the number of spikes each bin of the discharge interval (from 0–100 ms).(3)Auto-correlation analysis was defined as the number of neuron firing time differences (from −50 to 50 ms) in each time window.(4)LFP power was defined as the integral of the mean squared LFP. Here we took the average of multiple recording channels.

Statistical analyses and graphs were administrated by Origin Software and all results were expressed as mean ± SD unless otherwise specified. 

## 3. Results

### 3.1. Optical Stimulation Enhances Neuronal Activity

We conducted in vivo experiments with the integrated devices, first tested the response of neurons under optical stimulation and proved the performance of the MEA. In the region where the neurons successfully expressed ChR2, optical stimulation induced more neuronal activity. As shown in Figure 2a, the background noise of the electrode hardly changed before and during the illumination period. However, the spike firing rate significantly increased during the illumination period (Figure 2b). The LFP power also increased in the during-light period (Figure 2c). We also tested in non-viral transfection areas (as shown in Appendix A). It could be seen that the background noise had almost no change before and during the illumination, and the optical stimulation did not induce more spikes and increase lager LFP power, which proved that the application of light alone did not enhance neuronal activity. This indicated that the device successfully modulated the neuronal activity and measured the electrophysiology information, and these two experiments together illustrated the authenticity of optical stimulation-induced neuronal activity in the virus-infected area.

The experiment verified that at least in the range of brain depth 900–1250 μm, optical stimulation enhanced neuronal activity, indicating the excellent performance of the device in modulation and detection. As shown in Figure 3a, spikes significantly increased during optical stimulation at different test depths. Figure 3b shows the spike waveform extracted from the recording channels, and the results indicated that the spike waveforms measured at different depths were different. Due to the different optical stimulation pattern and opsin expression level at different test depths, the neuronal activity during illumination is not the same, but their neuronal activity was all significantly enhanced compared to before light illumination.

### 3.2. Optical Stimulation Activates a Possible Local Inhibitory Circuit

During the experiment, we attempted to analyze and study a local circuit combining optogenetics and electrophysiological recording. As shown in Figure 4a, the data was first recorded for 3 min without light, and then illumination (30 s light-on, 30 s light-off) lasting 6 min. Four neuron units were obtained (unit-a and unit-b were separated from the same recording channel) by principal component analysis and clustering from three recording channels. It can be seen from the figure that the spikes generated by unit-b,c,d increased significantly during optical stimulation. The spike firing rate of unit-b,c,d during optical stimulation was much greater than before optical stimulation (Figure 4b), which indicated that the neurons expressing ChR2 were successfully activated.

It was also found that the neuronal activity did not increase for unit-a, and even weakened during optical stimulation. In order to explore the reasons, we did a further analysis. Appendix A shows the auto-correlograms and ISI histograms of unit-a (Appendix A) and unit-b (Appendix A). The ISI histogram of unit-a (yellow) showed a short and sharp peak, and its auto-correlation analysis showed a decreasing trend, indicating that it had obvious cluster-like spike characteristics. For the unit-b (green) auto-correlation analysis, several equally spaced peaks appeared, indicating that it has periodic spike characteristics under optical stimulation. In terms of spike waveform, the amplitude of unit-a was almost twice that of unit-b (Figure 4c). Combined with their respective auto-correlation analysis, we speculated that unit-a may be a putative pyramidal neuron, while unit-b may be a putative interneuron [24]. Furthermore, we speculated that they were in a local circuit (dotted box in the Figure 4a), and the direction of synaptic transmission was from interneuron to pyramidal neuron, i.e., unit-b was the upstream neuron. The types of unit-c and unit-d were unclassified and their connection with others was not clear. When optical stimulation activated unit-b interneurons, its neuronal activity increased and its inhibitory effect on unit-a also enhanced. Therefore, spikes of unit-a were reduced during light. The LFP power (0–8 Hz) during illumination was lower than before and after illumination (Figure 4d), which also seems to prove our speculation. Appendix A showed the LFP power (0–30 Hz) before, during and after optical stimulation, which had the same trend as 0–8 Hz. Appendix A showed the power of LFP in different frequency bands. It was found that LFP was mainly concentrated in 0–8 Hz, so the subsequent LFP analysis was mainly concentrated at 0–8 Hz.

### 3.3. Effects of Different Optical Stimulation Patterns on Optogenetic Neuromodulation

The spikes of the three channels in different optical stimulation patterns were recorded (see Appendix A). According to the statistical average spike firing rate during optical stimulation, it was found that under the same duration, the spikes induced by s1 optical stimulation pattern were more than s2 optical stimulation pattern, the spikes induced by s2 optical stimulation pattern were more than s3 optical stimulation pattern (as shown in Figure 5a), the spike firing rate of the three channels had the same trend (Appendix A). On the whole, within a certain range, the lower the light frequency and the greater the duty cycle, the stronger the neuronal activity were caused. At the same time, we calculated the LFP power (0–8 Hz) under different optical stimulation patterns. The power under s1 was greater than that of s2 and s3, but there was no significant difference between s2 and s3 (Figure 5b). This may be due to the fact that LFP depended on thousands of neurons in a wide range, and there may be inhibition or excitation circuits in this area, which had complex effects on neuronal activity. In short, we believed that different optical stimulation patterns had an effect on the intensity of induced neuronal activity.

### 3.4. Effects of Different Light Powers on Optogenetic Neuromodulation

The output power of LED light source (P_s_) and the tip of optrode (P_t_) under different driving voltages is shown in Appendix A (unless otherwise specified, the output power refers to the output power of the optrode tip P_t_). We could see the maximum tip output power was 1.7 mW. We also simulated the power density at a certain distance from the fiber tip of the integrated device [13,25,26], see Appendix A for detailed process. Since the distance between the optical fiber tip and the recording site was 200 μm, the neurons were exposed to less than 10 mW/mm^2^ and the light would not cause great damage to the recorded neurons from the perspective of power density.

Then, the effect of different light power on neural activity was explored. We used 5000 mV, 4000 mV, 3000 mV, 2000 mV and 2500 mV to drive the laser one by one. The time of each stage was 2 min (after 40 s of illumination, the light was closed for 20 s, and the cycle was twice). According to the statistics of spike firing rate under different light power (Figure 6a), it was found that when the driving voltage of 2500 mV (corresponding to 1.032 mW output power) or above was used, the optical stimulation produced the effect of enhancing the activities of neurons; when the driving voltage of 2000 mV was used for optogenetic control, there was almost no obvious modulation effect. This showed that optogenetics required the light intensity to reach a certain threshold (1.032 mW in this study). At the same time, it was found that within a certain power range (1.032 mW–1.584 mW), the firing rate increased as the light power increased. When the driving voltage exceeds 4000 mV (corresponding to 1.584 mW), the firing rate decreased instead. On the one hand, it might be related to the switching property of photoreceptor itself, on the other hand, it might be due to the heat generated by light, resulting in the increase of neurons’ temperature, which might have a inhibition effect on the activity of neurons. There was also no obvious relationship between LFP power and light power (Figure 6b). 

## 4. Conclusions

In this paper, we employed a physical bonding method to integrate an optical fiber onto an MEA. This allowed us to characterize the activity of neural circuits in vivo, using optogenetic light delivery, and hence quantifying neural firing patterns. The changes of neuronal activity in a large range (depth from 900 to 1250 μm in the brain) under optogenetic control were detected and the infected neurons were successfully activated by optical stimulation, which proved the excellent performance of the integrated device. Furthermore, a possible circuit is analyzed by optogenetic modulation and electrophysiological detection, which provided a new method for circuit research. At the same time, we also studied the effect of different optical stimulation patterns and different light powers on the neuromodulation. We found that the optical stimulation patterns with lower frequency and larger duty cycle induced more intense neuronal activity. The light power needed to reach a certain response threshold (1.032 mW in this study), and greater optical power induced more intense neuronal activity within a certain power range.

In our work, the optical fibers are only used to transmit light. However, as far as we know, some new sensors based on optical fiber have great application potential in neuroscience research and attract many people’s attention. For example, researchers integrate high-resolution twist/torsion optical fiber sensors to monitor eventual movements of the patients utilizing the electromagnetic interference independency (EMI) of optical fibers [27]. Furthermore, some optical fiber-based sensors also are studied for MRI (Magnetic Resonance Imaging) interventions [28]. The optical fiber force sensors can help to provide force sensing in the MRI interventions, which increases the safety or accuracy of interventions. If the optical fiber on our integrated device is combined with photo sensor and customized circuit, it will have the potential to become an optical fiber force sensor. However, there are many other issues to consider in order to make it MRI-compatible. First of all, small fiber diameter and small footprint should be used to meet the requirement of sensor miniaturization. Secondly, we should choose the materials carefully. Polymer optical fiber can be considered for better flexibility and compatibility and the electrode can be based on polyimide, which matches brain tissue well in magnetic susceptibility. Finally, due to the potential of the electric current inducing imaging artifacts, some measures should be taken to prevent the interference of electrophysiology detection on MRI. In summary, our devices have the potential to be compatible with MRI. At the same time, we can also add in vivo electrochemical analysis function of neurotransmitters to the MEA. Many events in the nervous system are accompanied by the release of neurotransmitters, so the detection of neurotransmitters is particularly important. Electrochemical analysis need to modify multilayer polymers and enzymes at the electrode sites and the measurement is carried out in the three electrodes system, which are compatible with our devices [29]. Furthermore, it is also possible to measure the concentration of different neurotransmitters on an MEA by using different polymers or enzymes. These ideas will make it possible to simultaneously detect electrophysiological signals and neurotransmitters under the neuromodulation of optogenetics, which will undoubtedly promote the development of neuroscience. 

With the development of materials science, flexible electrodes have great advantages in the field of biosensors due to their close contact with brain tissue, many microelectrodes or optical devices based on flexible substrates have been studied [30,31]. The μ-LED array based on flexible substrate can accurately stimulate the brain at multiple points. Mccall and colleagues have made cellular-scale microscale, inorganic light-emitting diodes (μ-ILED, 6.45 μm thick, 50 × 50 μm^2^), and the μ-ILED is transferred to Polyethylene terephthalate (PET, polyester film) substrate for long-term implantation [32]. The thickness of organic light-emitting diodes (OLED) can even be less than 1 μm, which is more suitable for implantation in the brain [33]. Therefore, we can combine flexible electrode and optical interface to explore long-term optogenetic modulation and multi-mode sensor integration in the future [34,35]. In a word, the method proposed in this paper was an effective and feasible way of optogenetic neuromodulation and simultaneous neural detection, which can be applied to the study of neuroscience mechanism.

## Figures and Tables

**Figure 1 sensors-20-04526-f001:**
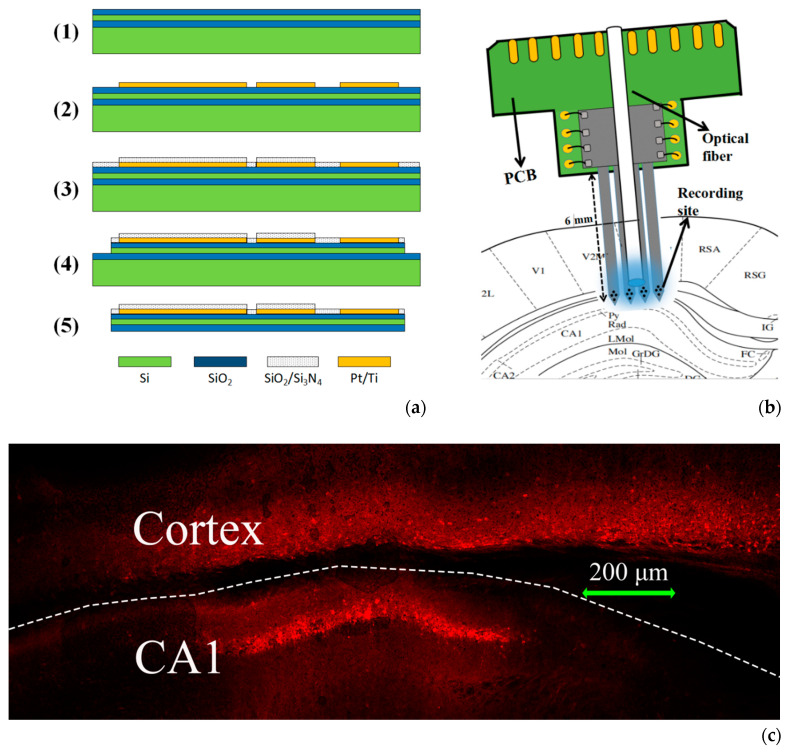
(**a**) Microelectrode array (MEA) manufacturing process: (1) Deposition of a layer of SiO_2_ on silicon-on-insulator (SOI). (2) Photolithography, sputtering and lift-off to form conductive layer. (3) Deposition of the insulating layer and exposure of recording sites and bonding pads. (4) Formation of the shape of the MEA. (5) Release of the MEA from the substrate. (**b**) Schematic diagram of the integrated device inserted into hippocampal CA1 of brain. (**c**) The fluorescence image of brain area infected by rAAV.

**Figure 2 sensors-20-04526-f002:**
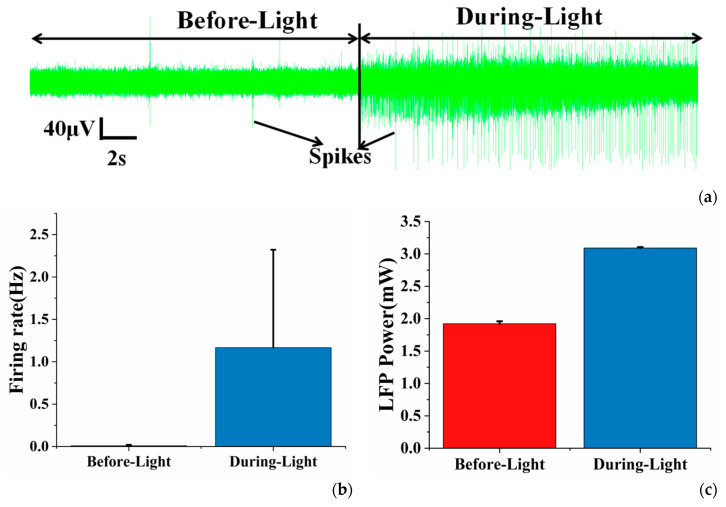
(**a**) The real-time recordings of electrophysiological signals before and during light illumination in viral transfection areas; (**b**) the average spike firing rate of neurons before and during optical stimulation; (**c**) the mean LFP power (0–30 Hz) of neurons before and during optical stimulation. Error bars indicate standard deviation of 3 channels.

**Figure 3 sensors-20-04526-f003:**
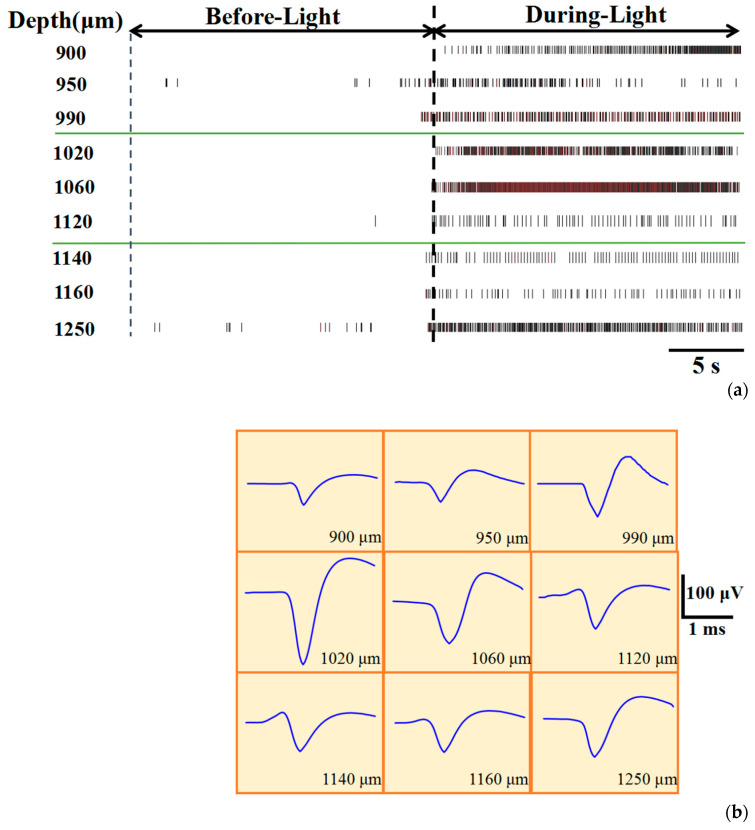
(**a**) The real-time recordings of spikes before and during blue light illumination at different depths (here we only selected one channel for each group of experiments as a display); (**b**) the spike waveform of neurons at different depths.

**Figure 4 sensors-20-04526-f004:**
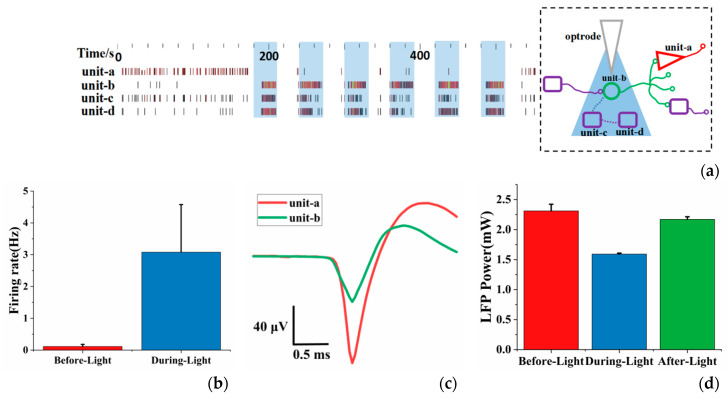
(**a**) The real-time recordings of spikes of 4 units separated from three recording channels (the shaded area is the period of during-light, 30sm) and inferred local circuit (the green circle represents the interneuron, the red triangle represents pyramidal neuron, the purple rectangles represent unknown type of neurons and dotted lines between neurons indicate uncertain connections); (**b**) the average spike firing rate of neurons before and during optical stimulation; (**c**) spike waveform of unit-a and unit-b; (**d**) the mean LFP power (0–8 Hz) of neurons before, during and after optical stimulation. Error bars indicate standard deviation of 3 channels and N = 6 (N is the number of experiments).

**Figure 5 sensors-20-04526-f005:**
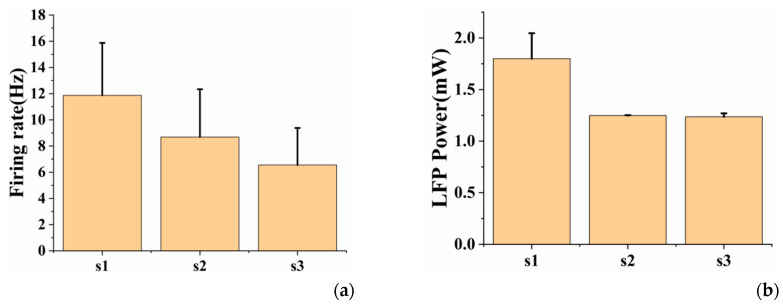
(**a**) The average spike firing rate of neurons under different optical stimulation patterns (s1: 10 Hz, duty ratio = 50%, 2 min; s2: 10 Hz, duty ratio = 25%, 2 min; s3: 16.6 Hz, duty ratio = 25%, 2 min); (**b**) The mean LFP power (0–8 Hz) of neurons under different optical stimulation patterns. Error bars indicate standard deviation of 3 channels.

**Figure 6 sensors-20-04526-f006:**
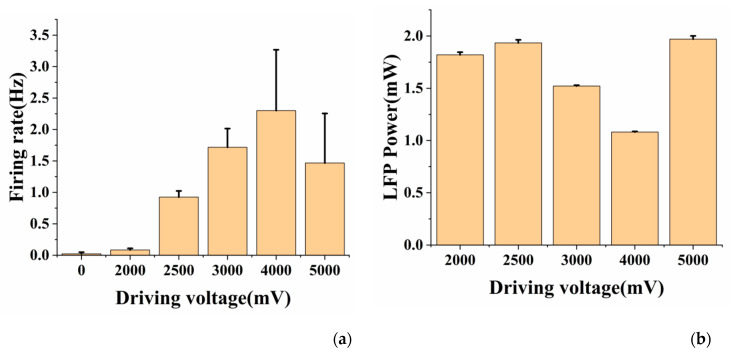
(**a**) The average spike firing rate of neurons under different driving voltages; (**b**) The mean LFP power (0–8 Hz) of neurons under different driving voltages. Error bars indicate standard deviation of 3 channels and N = 2 (N is the number of experiments).

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
