# Peer review of "In Vivo Optogenetic Modulation with Simultaneous Neural Detection Using Microelectrode Array Integrated with Optical Fiber"

_sensors, 2020, doi:10.3390/s20164526_

Round 1

Reviewer 1 Report

There is great interest in the development of improved tools that can simultaneously transmit light to targeted cells for optogenetic control and also detect electrophysiological activity concurrently from neurons.

In this study, the author’s used a physical bonding method to integrate an optical fiber onto a microelectrode array. This allowed them to combine optogenetics and electrophysiological recordings to obtain “neuromodulation” whilst recording “neuronal activity”. The rationale for the study is high. The approach could provide an efficient method for the detection and modulation of neurons in vivo and an effective tool to study neural circuits in the brain.

The technique presented is very interesting and potentially very useful. Other laboratories are also working on this, but this is novel. The author’s show proof of principle that they can stimulate in vivo and record increased neural activity.

I realise English is probably not the author’s first language, but there are a number of places where English needs to be improved. May I suggest having it read by a someone else proficient in English. I have tried to offer suggestions to improve English in certain places.

MAJOR POINTS:

It is not clear how many mice were used and how many experiments were performed. Please include N numbers for data measurements that are in the graphs and SEM in the results.

As the author’s acknowledge, heat is a problem with optogenetics. The fact that light did not activate regions of brain without ChR2 is good evidence that heat is not affecting their data. Do the authors have data to include showing the effects of a single pulse of light of varying intensities (e.g. 1mW and increasing ?). This would be good to include if they have it.

In 14 places the word “realized” is used. This is the wrong word. It does not read well. In all places where the word “realized” is used please replace it with something else appropriate. Perhaps, “revealed” or “determined” or “uncovered” at the appropriate place in the specific sentence. Obviously the word needs to be chose to fit the sentence.

I would like to see the image(s) of the fluorescence brain sections (stated to be in the supplementary figures - showing viral injections). This website came up with a URL error. Please resend the images and I suggest embed into the manuscript.

In fact, it would be good to see all the supplementary data and materials.

The first sentence of the discussion states “In this paper, physical bonding method was used to integrate the optical fiber on the MEA, which realized the function of in vivo optogenetic modulation and neural information detection.” This does not read well. I suggest changing this to “In this paper, we employed a physical bonding method to integrate an optical fiber onto an MEA. This allowed us to characterize the activity of neural circuits in vivo, using optogenetic light delivery, and hence quantifying neural firing patterns.”

Rather than say: “We realized the activation and electrophysiological detection of infected neurons at the depth range of 900 - 1250 μm in the brain which covers hippocampal CA1 and a part of the upper cortical area, analyzed a possible local inhibition circuit by combining opotogenetic modulation and electrophysiological characteristics and explored the effects of different optical patterns and light powers on the neuromodulation.”

In the introduction, please mention that optogenetics has been shown to be effective not just in the brain, but also the peripheral nervous systems. Please reference: Hibberd T et al. 2018 (see below) and Spencer NJ and Hu H. (2020) see below.

It was found that optogenetics combined with neural recording could provide more information and ideas for neural circuit recognition.

Line 14: Change “In this study, optical fiber and microelectrode array were” to “In this study, an optical fiber and a microelectrode array were..”

Line 43: replace “..the brain mechanisms..” with “…the neurophysiology of the central nervous system.,”

Line 54: This sentence needs modification. It is inappropriate to start a sentence mentioning the name of someone. “Jing Wang bonded a cannula to the open hole on the backside of the MEA to guide the fiber[13].” You could say “Wang and colleagues bonded a….”

Line 55: This sentence needs modification. “This method can usually achieve optogenetic control well, but need labor-intensive manual process.” I suggest “This method can be successfully applied to the field of optogenetics, but requires considerable time and effort to develop.”

Line 152: The figure legend 2: should say “The real-time recordings of electrophysiological signals,,” not “,.signal.”

Line 166: “spikes firing rate” should be “spike firing rate”

Line 195: Change “..we tried to..” to “…we attempted to..”

Line 282: did the authors mean to say “..flexible electrons

References that need to be cited in any revision:

Hibberd TJ, Feng J, Luo J, Yang P, Samineni VK, Gereau RW (4th), Kelly N, Hu H & Spencer NJ (2018) Optogenetic induction of colonic motility in mice. Gastroenterology May 18. pii: S0016-5085(18)34562-1. doi: 10.1053/j.gastro.2018.05.029.

Spencer NJ & Hu H (2019) The Enteric Nervous System. Sensory transduction, neural circuits and gastrointestinal motility. Nat Rev Gastroenterol Hepatol. 2020 Mar 9. doi: 10.1038/s41575-020-0271-2.

Author Response

Dear reviewer,

Thank you very much for your kindly suggestions and professional comments. Please see the attachment for detailed reply

Reviewer 2 Report

This manuscript proposes the integration of an optical fiber on a Microelectrode Array (MEA) for in vivo optogenetic modulation and neural information detection. This is a very interesting work for the research community, with a lot of potential to be further studied. The paper is very well structured, properly describing all the steps followed during the study. The supplementary material provided is very useful, featuring a brief set of mathematical foundations which I consider correct, to the best of my knowledge. The results of the paper are statistically validated and the figures are relevant and complement the results and explanations in a proper way. All in all, I consider this work as very good, but I recommend to perform the following set of minor but mandatory revisions, in order to slightly improve the quality of the paper before being published:

- The setup described in the paper consists of an optical fiber integrated on a MEA. Could the authors include a brief proposal of the necessary steps, or the materials required to make this setup MRI (Magnetic Resonance Imaging)-compatible? I think this could be a potential follow-up topic for research.

- Nowadays there is a huge interest on studying the potential use of fiber optics for neuroscience research. For example, in [IEEE Sensors Journal 17(7), 1952-1963 (2017)] some optical fiber-based sensors are studied for MRI interventions. In the present work, the authors propose a setup with a high potential for MRI uses. Additionally, the authors could also integrate high-resolution twist/torsion optical fiber sensors [JOSA B 36(5), 1176-1182 (2019)] in order to monitor eventual movements of the patients, thanks to the electromagnetic interference independency (EMI) of optical fibers. Could the authors add a paragraph or some comments providing a very brief state of the art of optical fiber sensors used for these purposes? Please, include the two aforementioned references as well.

- Please, slightly increase the font size of the bottom part of Figure 1b, since the labels are barely readable.

- Line 96-97: Could you provide some extra details regarding the optical fiber used in this work? Is it a silica or polymer optical fiber (POF)? According to the diameter it might be multimode. Can you confirm this assumption?

Author Response

(The authors gave the same response as above.)
